# Local Weather Conditions Create Structural Differences between Shallow Firn Columns at Summit, Greenland and WAIS Divide, Antarctica

**Ian E. McDowell [1],\*** , **Mary R. Albert [2]**, **Stephanie A. Lieblappen [2]** and **Kaitlin M. Keegan [1]**

1   Graduate Program of Hydrologic Sciences and Department of Geological Sciences and Engineering, University of Nevada, Reno, 1664 N. Virginia St., Reno, NV 89557, USA; kkeegan@unr.edu
2   Thayer School of Engineering, Dartmouth College, 14 Engineering Dr., Hanover, NH 03755, USA; Mary.R.Albert@dartmouth.edu (M.R.A.); steph.lieblappen@gmail.com (S.A.L.)
\*   Correspondence: ian.mcdowell@nevada.unr.edu

**Abstract:** Understanding how physical characteristics of polar firn vary with depth assists in interpreting paleoclimate records and predicting meltwater infiltration and storage in the firn column. Spatial heterogeneities in firn structure arise from variable surface climate conditions that create differences in firn grain growth and packing arrangements. Commonly, estimates of how these properties change with depth are made by modeling profiles using long-term estimates of air temperature and accumulation rate. Here, we compare surface meteorology and firn density and permeability in the depth range of 3.5–11 m of the firn column from cores collected at Summit, Greenland and WAIS Divide, Antarctica, two sites with the same average accumulation rate and mean annual air temperature. We show that firn at WAIS Divide is consistently denser than firn at Summit. However, the difference in bulk permeability of the two profiles is less statistically significant. We argue that differences in local weather conditions, such as mean summer temperatures, daily temperature variations, and yearly wind speeds, create the density discrepancies. Our results are consistent with previous results showing density is not a good indicator of firn permeability within the shallow firn column. Future modeling efforts should account for these weather variables when estimating firn structure with depth.

**Keywords:** firn density; firn permeability; polar meteorology; Greenland ice sheet; West Antarctic ice sheet

## 1. Introduction

Firn is perennially persistent snow on glaciers and ice sheets that densifies over time and eventually transitions to glacial ice. Firn exists within the upper portion of the ice column in accumulation zones, which are spatially extensive on ice sheets. For example, based on identification of the snowline in Greenland using remote sensing [1], firn covers over ~85% of the ice sheet. Additionally, maximum firn depths can reach ~120 m in the interior of Antarctica [2]. Firn structure is critically important to our understanding of three active areas of research: (1) Correctly interpreting gas concentrations contained within sealed bubbles from ice cores as climate proxies (e.g., [3–5]); (2) estimating the buffering capacity of firn on sea level rise as the areal extent of melt increases on ice sheets by understanding the interplay between meltwater infiltration and firn densification (e.g., [6–8]); and (3) determining ice sheet mass balance using satellite altimetry to estimate ice sheet volumetric changes from recorded changes in surface height if the density is well known (e.g., [9–11]).

A common approach to estimating firn densification at a location with depth is to model the depth-density relationship by relating overburden pressure and temperature to the densification rate

using long-term averages of site mean annual air temperature and accumulation rate [12]. Whereas there are many firn densification models with different governing equations, these models all generate steady-state and transient density estimates by simulating changes to the temperature and accumulation rate boundary conditions [13]. Using parameters derived from density, such as open porosity, other empirical models can be used to estimate other properties such as firn permeability and gas diffusivity within the firn column [14,15], although studies have shown that these variables are not well estimated using firn density. The semi-empirical model by Herron and Langway (1980) [12] is useful for determining the subsurface properties of firn at depths near the pore close-off range and at sites lacking more detailed meteorological information. However, given the influence of shallow firn on gas-mixing, meltwater infiltration, and compaction-driven changes in the ice sheet surface, the ability to accurately model the shallow firn structure is needed.

Here, we present density and permeability measurements from two shallow firn cores collected at sites that are characterized by the same mean annual air temperatures and accumulation rates: Summit Station in Greenland, and the West Antarctic Ice Sheet (WAIS) Divide in Antarctica. Given the similarities in long-term meteorological conditions, WAIS Divide was selected to serve as an Antarctic ice core site analogous to Summit, Greenland, where the GISP2 drilling project occurred. The mean annual air temperature is approximately −30 °C at Summit and WAIS Divide, with an annual snow accumulation rate of approximately 0.22 m yr$^{-1}$ [16,17]. Given that the Herron and Langway (1980) [12] firn density model serves as a standard in firn densification modeling and that many other models have originated as derivations of this benchmark model, we compare the density profiles to a model profile generated from the Herron and Langway empirical firn densification model in order to test the inherent assumptions. This model would predict the same modeled density output given that the model inputs are the same. We compare the permeability profiles to the models of Freitag et al. (2002) [15] and Adolph and Albert (2014) [14], which are empirical models for the permeability profile based on the relationship between permeability and open porosity that can be determined by firn density. Using additional meteorological data from automated weather stations nearby the coring sites, we discuss potential factors that create discrepancies in the observed and modeled firn density and permeability profiles and highlight their importance for future firn modeling.

## 2. Materials and Methods

### 2.1. Firn Core Collection

In this study, we compare the physical properties of 2 shallow firn cores, 1 drilled at Summit Station, Greenland (elevation ~3200 m) and the other at the West Antarctic Ice Sheet Divide (WAIS Divide) site (elevation ~1800 m) (Figure 1). These 2 sites provide high-resolution density and permeability data while also providing long-term meteorological data from Automated Weather Stations (AWS) located nearby the firn core collection site. The Summit firn core (SUFA07) was retrieved in the summer of 2007, while the WAIS Divide firn core (WDC05) was retrieved during the austral summer of 2005. Both cores were drilled to the pore close-off depth, packed on-site into insulated boxes, and shipped to the US Army Cold Regions Research and Engineering Laboratory (CRREL) in Hanover, NH, USA. In the laboratory, the firn samples were stored at −29 °C in a cold room.

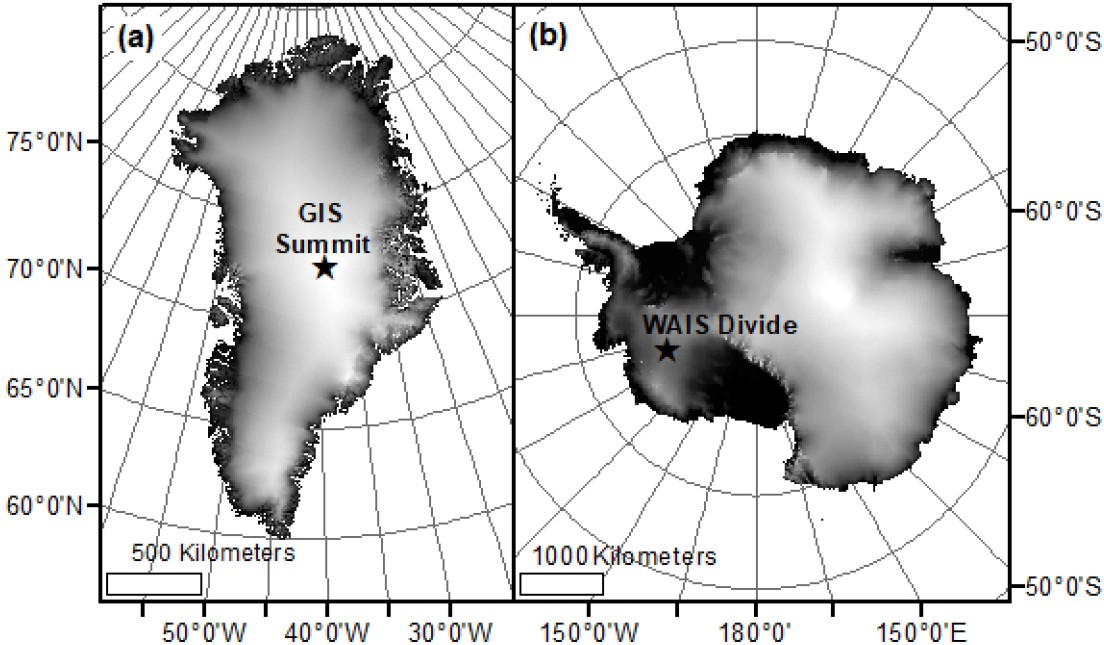

**Figure 1.** Location of firn core collection and automated weather station sites: (**a**) Summit, Greenland (GIS Summit) and (**b**) West Antarctic Ice Sheet (WAIS) Divide, Antarctica (WAIS Divide). Coring locations are plotted on the Greenland 5 km digital elevation model (DEM) [18] and the Antarctic 1 km DEM [19]. Greenland and Antarctica are displayed using the WGS 1984 Arctic and Antarctic Polar Stereographic Coordinate systems, respectively. The Summit firn core was collected from ~3200 m above sea level and the elevation at WAIS Divide was ~1800 m.

### 2.2. Density and Permeability Measurements

Both the Summit and WAIS Divide firn cores were divided into subsamples of 5–10 cm by depth depending on the grain sizes of the firn layers. The density of each sample was calculated by measuring the height and diameter using digital calipers (±0.05 mm), as well as the mass with a digital scale (±0.005 g). The permeability of these samples was measured using a custom permeameter developed for snow and firn samples [20]. After conducting a study of repeat measurements, the permeability was determined to have measurement error within 10%. Measurements of the density and permeability reported here were made between 2007–2012.

### 2.3. Meterological Data Processing

We used meteorological data collected from the AWS located at both Summit and WAIS Divide to determine average meteorological conditions at each site. The AWS at Summit is a part of the Greenland Climate Network [21], while the University of Wisconsin-Madison maintains the Kominko-Slade AWS at WAIS Divide. The downloaded meteorological dataset from Summit spanned 1 January 1997–31 December 2006. Unfortunately, the Kominko-Slade weather station was installed in 2006, after the shallow core from WAIS Divide had been collected. To analyze meteorological data over the same amount of time as the Summit data, we examined WAIS Divide meteorological data over the time period spanning 1 February 2009–31 December 2018, as data collected in the first 2 years after installation were characterized by relatively large amounts of noise. Although we acknowledge the limitation of not having the local meteorological information that influences firn deposition at WAIS Divide, we do not believe that it greatly affects our analysis. To verify this claim, we examined NCEP-NCAR reanalysis temperature and relative humidity data spanning the years 1995–2019 [22] (Appendix A). At WAIS Divide, temperature and relative humidity were relatively constant, changing by 0.006 °C yr$^{-1}$ and −0.027% yr$^{-1}$, respectively, as determined by a least-squares regression fit to

the reanalysis data. These small trends suggest that, although we analyzed AWS data from a 10-year period after the WAIS Divide firn core was collected, we examined meteorological conditions that were similar to those that influenced snow deposition and firn metamorphism in the core.

We limited our analysis to 3 variables, all of which have been shown to drive snow metamorphism: Air temperature, relative humidity, and wind speed. The AWS at WAIS Divide only provided air temperature, relative humidity, wind speed, wind direction, and air pressure. Thus, to examine the same variables at each site, we focused our analysis on the 3 aforementioned characteristics. The downloaded data output consisted of hourly averages of 1 min measurements at Summit and 10 min averages of 30 s measurements at WAIS Divide, resulting in datasets that were approximately 8800 and 52,000 entries long, respectively, for each year. The relative humidity values from Summit were reported in terms of saturation vapor pressure with respect to ice. We converted the relative humidity data downloaded from the AWS at WAIS Divide by multiplying by the ratio between the saturated vapor pressures with respect to water and with respect to ice at the corresponding temperature recorded with the relative humidity value. Relative humidity measurements at WAIS Divide often oscillated between 0% and 100%. We filtered these data by eliminating all measurements that fell outside ±3 standard deviations from the mean, which should include >99% of natural variability in the data. Additionally, the maximum recorded windspeed at WAIS Divide reached 32 m s$^{-1}$ in 2016 according to the UW Madison AWS index website, so all reported values above this maximum speed were removed before conducting our analysis.

### 2.4. Determining Bulk Firn Properties and Mean Meteorological Conditions

We compared firn properties across depths where there were measurements from both Summit and WAIS Divide sites. The fragile upper meters of the WAIS Divide firn precluded measuring physical properties until depths at which the firn core was sufficiently sintered to survive transport, starting at approximately 2 m depth. Because we only had firn permeability measurements from the WAIS Divide core between ~3.5 m and 11 m, we compared both measured density and permeability values over 3.5 m to 11 m depth for consistency. In addition to examining the profiles of the properties with depth, we also compared the median bulk density and permeability over 3.5 m to 11 m at the 2 sites due to the non-Gaussian distribution of Summit permeability values.

For meteorological variables of interest at each site, we merged data collected on the same Julian day of the year from each of the years in the record. Because we examined measurements collected at different intervals (10-min vs. 1-h), we calculated hourly averages of each meteorological variable at WAIS Divide and daily means at both sites, along with the associated variance and standard deviation, so that we could compare meteorological variables over the same standardized period of time. We determined the mean meteorological value over the 10-year time periods of interest by averaging the daily means and calculating the pooled standard deviation. Additionally, to examine seasonal variations in temperature, we calculated the mean summer and winter temperatures and associated standard deviations for the two sites. We defined summer temperatures for Summit and winter temperatures for WAIS Divide as those that were recorded between June 1 and August 31 of each year, and we considered summer temperatures at WAIS Divide and winter temperatures at Summit to fall between December 1 and February 28. Finally, since large daily temperature differences are conducive to hoar formation, we generated a yearly record consisting of averages from each measurement interval over the meteorological record and determined the daily maximum and minimum of this averaged yearly record for both locations. All meteorological values reported in the manuscript are presented as the mean ± the standard deviation.

### 2.5. Modeled Density and Permeability Profiles

To examine how well the commonly used Herron and Langway firn density model reproduces the firn density-depth relationship at Summit and WAIS Divide, we compared the measured density profiles with an idealized modeled profile based on the Herron and Langway empirical model for

the first stage of densities less than 0.55 kg m$^{-3}$ [12]. The model relies on the site accumulation rate, mean annual air temperature, and initial surface snow density. We generated modeled profiles using surface snow densities of 250 kg m$^{-3}$, 340 kg m$^{-3}$, and 380 kg m$^{-3}$ based on the range of observations of surface snow densities from Summit and WAIS Divide. Schwander et al. (1993) [23] reported a surface snow density of 340 kg m$^{-3}$. However, the lowest seasonal surface density has been measured at 250 kg m$^{-3}$ [24]. At WAIS Divide, Fegyveresi et al. (2018) [25] observed a maximum surface density of 380 kg m$^{-3}$. We varied these surface snow density boundary conditions as a method to estimate the uncertainty of the modeled density and permeability profiles. We then used the measured site mean annual air temperature and accumulation rate, and generated a synthetic density profile for Summit and WAIS Divide to examine how well the model could replicate observed density changes with depth.

Site-specific power law relationships with open porosity have been identified [14,15] based on open porosity estimated from firn density. Freitag et al. (2002) applied an experimental and modeling technique to determine firn permeability from distinct depths of a core drilled in Greenland to develop an empirical power law relationship between open porosity and permeability [15].

$$k \; = \; 10^{-7.7} \; m^2 \; \phi_O^{3.4} \tag{1}$$

where $k$ is permeability, $\phi_o$ is open porosity, and $m$ is a material constant that has been experimentally determined to be 1.5 for firn. Open porosity can be calculated by subtracting closed porosity, $\phi_c$, from total porosity, $\phi$, where closed porosity is determined by an exponential relationship between the firn density at depth and the critical pore close-off density.

$$\phi_C = \begin{cases} \phi \, e^{\left[75*\left(\frac{\rho(z)}{\rho_{co}}\right)-1\right]}, & 0 \; < \; \rho(z) \; < \; \rho_{co} \\ \phi, & \rho_{co} \; \leq \; \rho(z) \end{cases} \tag{2}$$

where close-off density is represented by $\rho_{co}$ and $\rho(z)$ is the firn density at a given depth within the firn column.

Using data that spanned the entire firn column, Adolph and Albert (2014) [14] improved the power law relationship:

$$k \; = \; 10^{-7.9} \; m^2 \; \phi_O^{3.71} \tag{3}$$

The authors showed that, even when using measured density and permeability from the same site [14], the power law relationship overpredicts the permeability for most of the range considered, and because there is much more natural variability in firn structure than porosity alone can describe, power law relationships are not able to adequately replicate the permeability profile. Density describes the mass per unit volume of firn, but the packing arrangement within the unit volume, which affects firn permeability, can vary. Thus, density, and therefore open porosity, is unable to capture microstructural aspects that have a large effect on permeability. Nevertheless, here we used Equations (1) and (3) from Freitag et al. (2002) [15] and Adolph and Albert (2014) [14] to generate modeled permeability profiles based on our modeled firn density profile.

*2.6. Statistical Analyses*

We compared differences between firn characteristics and meteorological variables using 2-tailed t-tests with pooled variances. Several of these variables were only marginally normally distributed (as assessed via probability plots). Therefore, we log$_{10}$-transformed all density and permeability datasets prior to conducting statistical comparisons. We assessed significance at $\alpha = 0.05$.

Because the meteorological datasets were so large, a t-test to compare mean values resulted in very large t-values and extremely low *p*-values. To mitigate the chances of reporting a statistical significance solely driven by the large dataset, the mean annual values reported were averages of the daily mean values and standard deviation was calculated by pooling the daily variances. This allowed us to conduct t-tests on fewer measurements. Additionally, we chose to also evaluate and report the

effect size for all comparisons. Effect size measures the difference between groups and is commonly represented by the value Cohen's *d* [26]. This value is determined by finding the difference between the 2 means being compared and divided by the pooled standard deviation.

$$d = \frac{|\mu_1 - \mu_2|}{\sqrt{SD_1^2 + SD_2^2}} \qquad (4)$$

In this equation, $\mu$ represents the sample means of groups 1 and 2 and *SD* is the standard deviation of both groups. Generally, if *d* is greater than 0.8, the effect size is considered to be large enough to conclude that a difference exists between the 2 groups being compared, while values between 0.5–0.8 suggest there is potentially a statistical difference between the 2 sample means, and a value less than 0.2 indicates that there is not a significant difference between the two groups [26]. Effect sizes, along with the calculated *p* values for the differences in the mean firn characteristics and meteorological conditions, are listed in Table 1.

**Table 1.** Meteorological characteristics and firn properties of Summit and WAIS Divide.

| | GIS Summit | WAIS Divide | *p* Value of Mean Difference | Effect Size (*d*) |
|---|---|---|---|---|
| *Meteorological characteristics* | | | | |
| Annual air temperature (°C) | −29.0 ± 8.2 | −28.3 ± 7.2 | 0.28 | 0.10 |
| Summer air temperature (°C) | −14.2 ± 5.0 | −17.7 ± 4.1 | <0.001 | 0.77 |
| Winter air temperature (°C) | −39.8 ± 10.1 | −35.8 ± 3.4 | <0.001 | 0.52 |
| Annual relative humidity (%) | 91.8 ± 10.0 | 93.4 ± 6.3 | <0.001 | 0.20 |
| Annual wind speed (m s$^{-1}$) | 4.0 ± 3.0 | 6.6 ± 4.0 | <0.001 | 0.74 |
| Annual accumulation rate (m yr$^{-1}$) | 0.22 [16] | 0.22 [17] | - | - |
| *Firn properties* | | | | |
| Firn density * (kg m$^{-3}$) | 0.447 ± 0.048 | 0.494 ± 0.038 | <0.001 | 1.16 |
| Firn permeability * (× 10$^{-10}$ m$^2$) | 35.2 ± 26.5 | 33.9 ± 22.5 | 0.01 | 0.36 |

Values reported as mean ± standard deviation. * Median value between 3.5–11 m.

## 3. Results

### 3.1. Firn Properties

Firn density generally increases with depth. However, permeability has a more complex relationship with depth. Albert and Shultz (2002) [27] showed that measured permeability profiles have interannual variations, but in general, at Summit, the lowest permeability in the top 20 m occurs in the wind pack at the surface. Below the wind pack, the permeability increases with depth until approximately 3 m due to firn metamorphism before decreasing with depth below 3 m.

The firn density profiles from Summit and WAIS Divide can be seen in Figure 2. Whereas modeling of interstitial processes should be done using firn property profiles that vary with depth, assessment of grouped measurements is useful in interpreting firn structure using geophysical or remotely sensed measurements. Grouping individual measurements into histograms show that Summit firn was less dense, with most density measurements falling between 0.3 kg m$^{-3}$ and 0.5 kg m$^{-3}$ (Figure 3). The authors note that the Summit permeability data followed a non-normal distribution, and therefore, summary statistics should be interpreted with caution. Due to the non-Gaussian nature of the Summit firn permeability, we reported all firn density and permeability values as the median ± the standard deviation. However, we did report *p* values and effect sizes that were calculated using the log$_{10}$-transformed mean values. Firn within the upper 3.5 m to 11 m of the firn column at Summit had a median density of 0.447 ± 0.047 kg m$^{-3}$, while firn within the same depths at WAIS Divide had a median density of 0.484 ± 0.038 kg m$^{-3}$. The difference between the mean firn densities at both

sites were statistically significant (*p* value < 0.001) when the means were compared using a t-test with pooled variances.

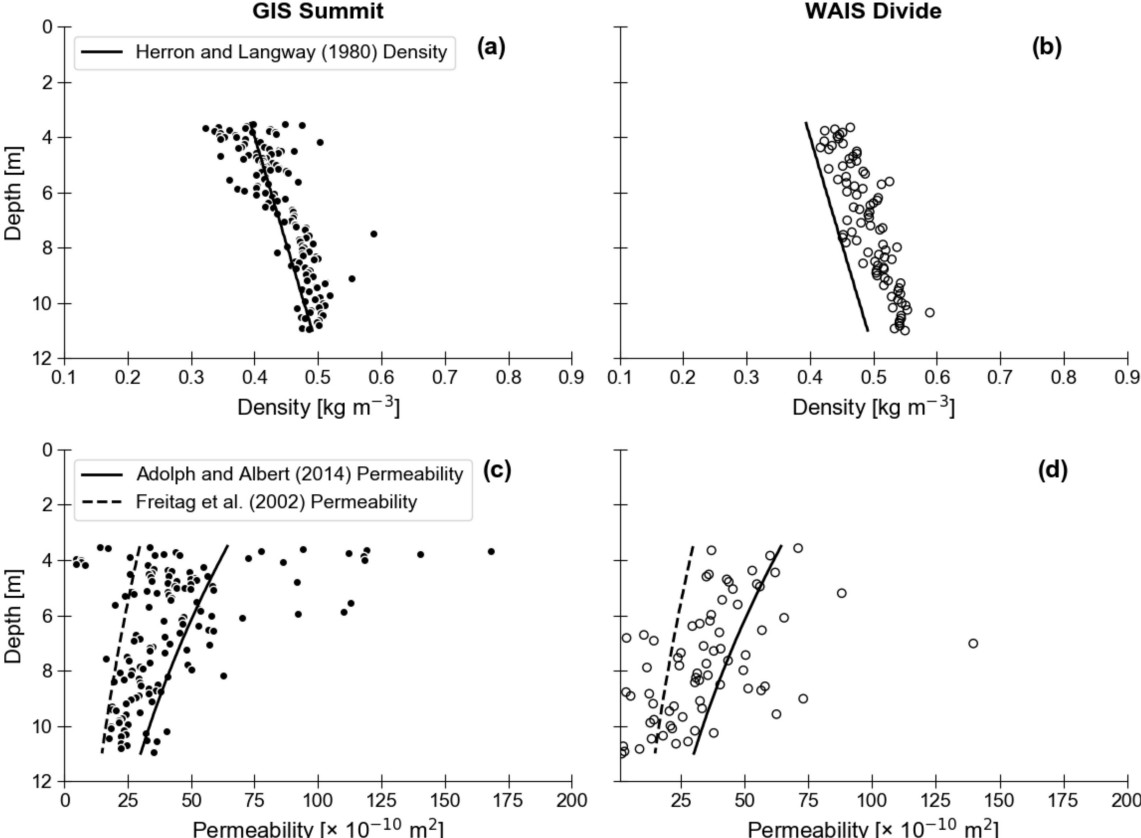

**Figure 2.** Firn density profiles (**a,b**) and permeability profiles (**c,d**) between 3.5 m and 11 m depth at GIS Summit (**a,c**) and WAIS Divide (**b,d**). The solid black line in panels (**a,b**) is the modeled firn density profile generated from Herron and Langway (1980) [12]. In panels (**c,d**), the solid black line represents the modeled permeability profile from Adolph and Albert (2014) [14] and the dashed black line shows modeled permeability profiles using the model of Freitag et al. (2002) [15]. All modeled density and permeability profiles shown were generated using an accumulation rate of 0.22 m yr$^{-1}$, mean air temperature of −28 °C, and surface snow density of 0.34 kg m$^{-3}$.

The modeled density and permeability profiles in Figure 2 were generated using a surface snow density of 0.34 kg m$^{-3}$. Profiles generated using surface densities ranging from 0.25–0.38 kg m$^{-3}$ varied by less than 0.1% at each depth. Thus, these profiles are not shown.

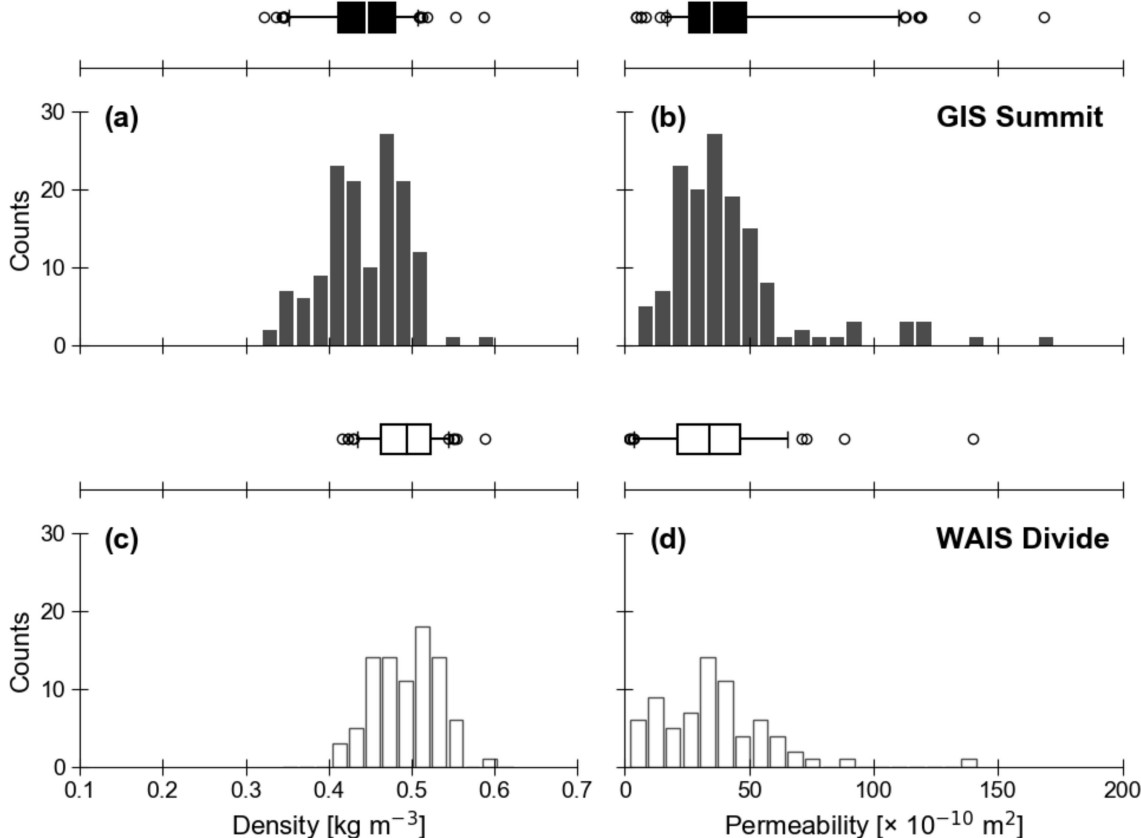

**Figure 3.** Distribution of density (**a**,**c**) and permeability (**b**,**d**) in the depth range of 3.5–11 m in the firn column at GIS Summit (**a**,**b**) and WAIS Divide (**c**,**d**). The number of measurements, *n*, used to create each histogram are as follows: GIS Summit density $n = 137$, GIS Summit permeability $n = 137$, WAIS Divide density $n = 86$, WAIS Divide permeability $n = 71$. Bin widths were set to 0.2 kg m$^{-3}$ for density histograms and $7.0 \times 10^{-10}$ m$^2$ for permeability histograms. Box plots above each distribution show interquartile range and whiskers extend to encompass 95% of the data. The median value is indicated by a line within the interquartile range.

Firn permeability profiles at depths between 3.5 m and 11 m generally appeared similar between the two sites (Figure 2). Between 3.5 m and 11 m depth, median firn permeability values at Summit and WAIS Divide were $35.2 \pm 26.5 \times 10^{-10}$ m$^2$ and $33.9 \pm 22.5 \times 10^{-10}$ m$^2$, respectively. Both median density and permeability values are reported in Table 1 for the two sites along with calculated *p* values and effect sizes of the log$_{10}$-transformed data. Whereas the permeability values were statistically different, with a *p* value of 0.01, the calculated effect size, 0.36, was rather small, so the difference in permeability between the two sites is likely trivial. Therefore, we conclude that, for depths between 3.5 m and 11 m, Summit firn is statistically less dense and likely similarly permeable on average than the firn at WAIS Divide.

*3.2. Site Meteorological Conditions*

The AWS data available from Summit overlap with the years of snow accumulation. However, meteorological data from WAIS Divide were collected after the firn core retrieval. Despite this discrepancy, analysis of NCEP-NCAR reanalysis data for WAIS Divide showed minimal changes to air temperature and relative humidity over the last 25 years at this site. Therefore, we assume that the analysis of the 10-year dataset, from a period after firn core collection at WAIS Divide, to be representative of the conditions influencing surface snow formation and metamorphism as these firn layers were formed.

The daily mean air temperatures at Summit and WAIS Divide varied seasonally, and the relative humidity daily means also appeared to show slight seasonal differences, particularly at WAIS Divide, where winter and summer relative humidity values differed by ~9%. However, wind speed and relative humidity appeared to be less affected by seasonality than temperature (Figure 4). Air temperatures at Summit averaged −29.0 ± 8.2 °C, while the mean annual temperature at WAIS Divide was −28.3 ± 7.2 °C. Despite having similar mean air temperatures, Summit had greater differences between mean summer and winter air temperature. Mean daily air temperatures at Summit were at a maximum of -10.8 °C in August and a minimum of −46.9 °C in February. Temperatures at WAIS Divide peaked in December at −8.1 °C and were lowest in June at −43.0 °C. The average summer temperature was −14.2 ± 5.0 °C at Summit and −17.7 ± 4.1 °C at WAIS Divide, while the average winter temperatures were −39.8 ± 10.1 °C and −35.8 ± 3.4 °C, respectively. Daily temperatures fluctuated daily by approximately 6.1 °C at Summit and 4.1 °C WAIS Divide, and Summit additionally experienced the greatest daily temperature variability observed at both sites during the spring (Figure 5).

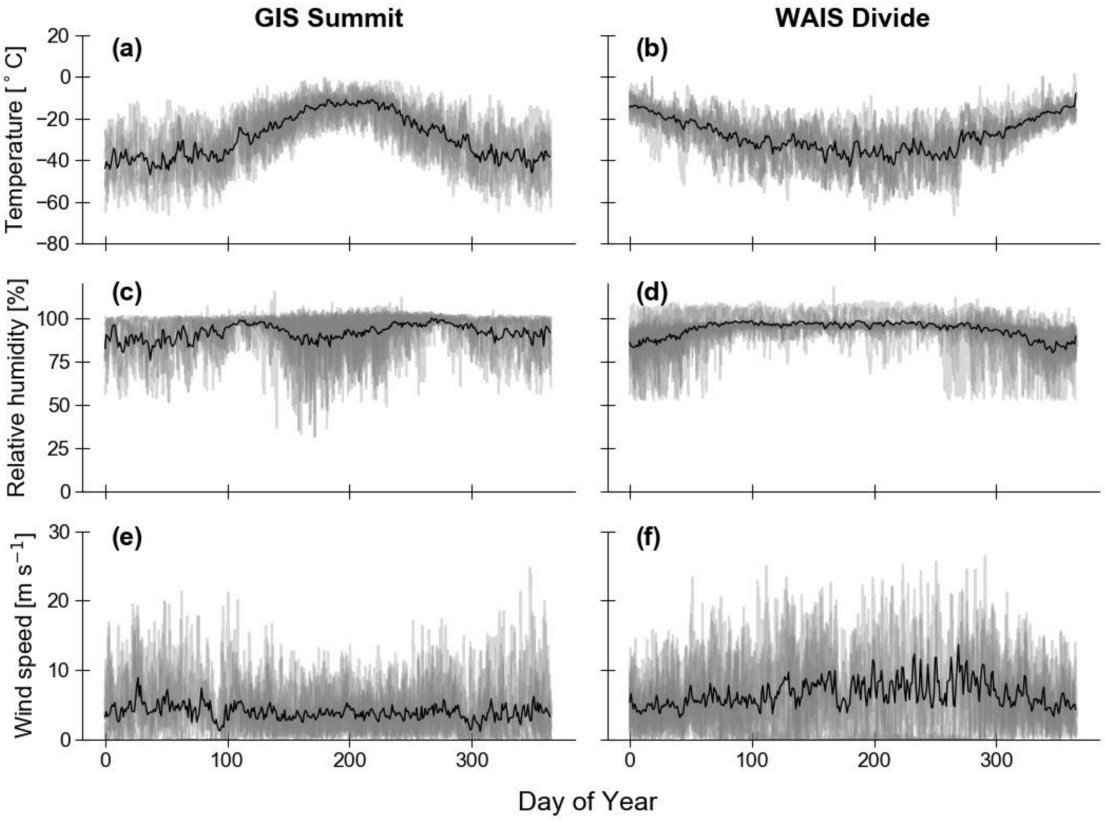

**Figure 4.** Meteorological data from both Summit (panels (**a**,**c**,**e**)) and WAIS Divide (panels (**b**,**d**,**f**). Air temperatures at the two sites are shown in panels (**a**,**b**), relative humidity values are displayed in panels (**c**,**d**), and wind speeds are plotted in panels (**e**,**f**). Hourly data are plotted from the years 1997–2007 for Summit and 2009–2019 for WAIS Divide vs. the day of the collection year with grey transparent lines, so that progressively darker greys show overlapping data. Black lines show a mean meteorological value for the given day of the year, averaged over the collection period.

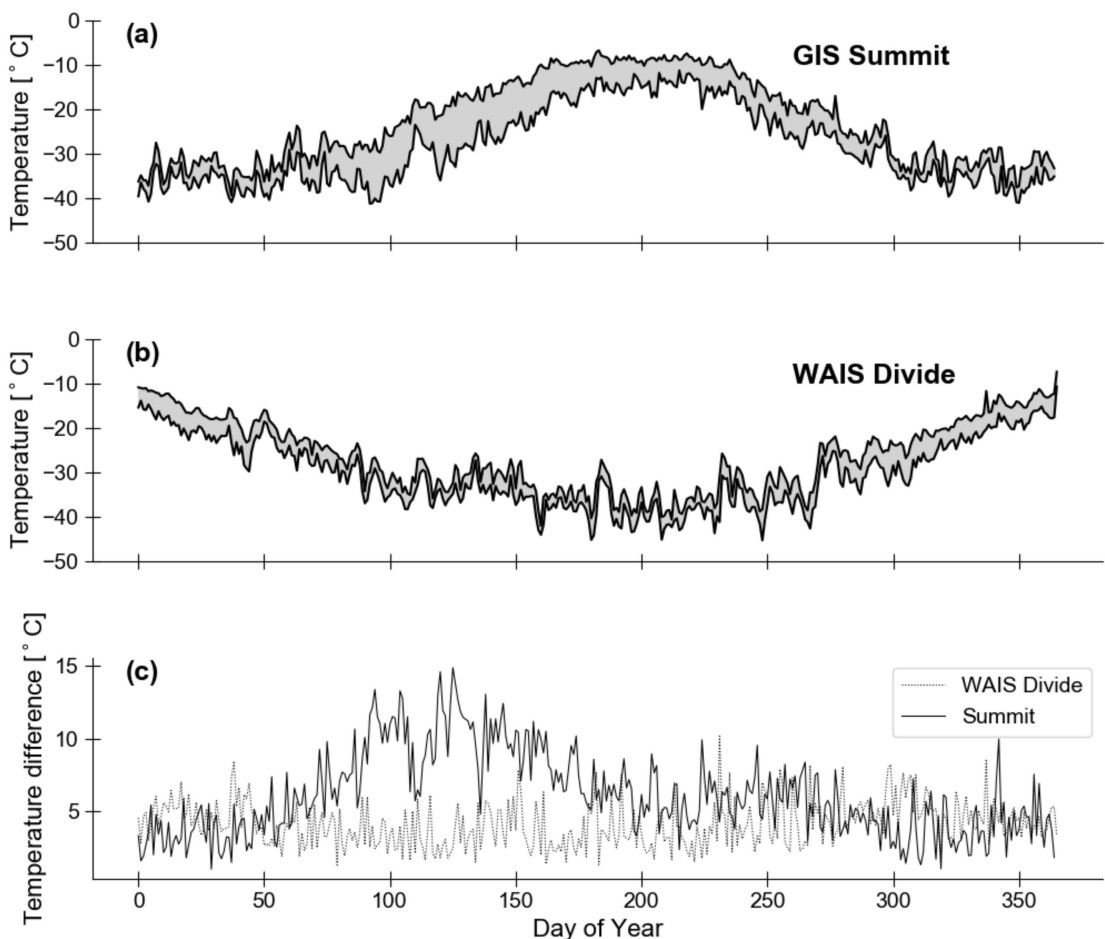

**Figure 5.** Variability of daily air temperatures at Summit (**a**) and WAIS Divide (**b**). Averaged daily minimum and maximum temperatures are plotted, with the shaded area representing the daily temperature variability. The difference between the highest and lowest daily temperatures are plotted in panel (**c**).

WAIS Divide has higher relative humidity values (93.4% at WAIS Divide compared to 91.8% at Summit). Relative humidity was calculated in terms of saturation vapor pressure over ice. Thus, it is possible that the values would have exceeded 100% if the air was saturated with super-cooled water vapor. WAIS Divide was windier on average, with a mean annual wind speed of 6.6 ± 4.0 m s$^{-1}$ compared to 4.0 ± 3.0 m s$^{-1}$ at Summit. Direct comparison of measured net shortwave radiation between the two sites is not possible with the data downloaded from the AWS. However, Scott and others (2017) [28] compared the summer mean net shortwave radiation between Summit and WAIS Divide and found a difference of less than 5 W m$^{-2}$, with a mean summer value of 59.25 W m$^{-2}$ and 55 W m$^{-2}$ at WAIS Divide and Summit, respectively. A comparison of site-specific meteorological characteristics can be found in Table 1.

Mean annual air temperatures at Summit and WAIS Divide were statistically similar ($p = 0.28$, $d = 0.10$). However, when comparing mean summer and winter temperatures, both variables were statistically different between the two sites ($p < 0.001$), with summer temperatures having an effect size of $d = 0.77$ and winter temperatures having a medium effect size ($d = 0.52$). Whereas a t-test suggests that the difference in relative humidity between the sites was significant ($p < 0.001$), the calculated effect size was small (0.20). Thus, relative humidity values were likely similar between the two sites. WAIS Divide had significantly higher wind speeds ($p < 0.001$; $d = 0.74$). Therefore, despite having mean annual temperatures that were not statistically different, Summit was characterized by greater seasonal temperature variability and lower wind speeds than WAIS Divide.

## 4. Discussion

We found that firn density differed significantly between Summit and WAIS Divide between 3.5–11 m depth in the firn column. At WAIS Divide, firn was denser at all depths between 3.5 m and 11 m within the upper firn column (Figure 2). Firn at Summit between 3.5 m and 11 m had a mean density of 0.447 ± 0.047 kg m$^{-3}$, while the median density between 3.5–11 m of the firn column at WAIS Divide was 0.494 ± 0.038 kg m$^{-3}$ (Figure 3; Table 1). Despite large density differences, the difference in average firn permeability between 3.5–11 m depth was statistically significant at a level of 0.01. However, there was a small effect size ($d = 0.36$), suggesting that there was likely a negligible difference in the bulk permeability between the two sites. We observed that the median permeability of firn at depths between 3.5 m and 11 m at Summit was 35.2 ± 26.5 × 10$^{-10}$ m$^2$ and 33.9 ± 22.5 × 10$^{-10}$ m$^2$ at WAIS Divide (Figure 2; Table 1). Our analysis of the AWS data demonstrated that both sites experienced the same mean annual air temperatures and snow accumulation (Table 1). However, summer temperatures at Summit were greater than at WAIS Divide and showed the greatest diurnal variability during the spring. Relative humidity was slightly higher at WAIS Divide (93.4%) than at Summit (91.8%). However, because the effect size was small, the difference in mean annual relative humidity was likely not statistically significant. Additionally, at WAIS Divide, windier conditions dominated, with a higher mean annual wind speed (Table 1) and more-frequent transient high wind speed events (Figure 4). These results suggest that differences in the average weather conditions at each site impacted the resulting firn column density profile.

Between depths of 3.5 m and 11 m, firn at WAIS Divide was consistently denser than firn at Summit and mean densities were significantly different despite both sites having similar mean annual air temperatures and accumulation rates. Based on the empirical firn densification model by Herron and Langway that determines firn density profiles given the surface snow density [12], mean annual air temperature, and average annual accumulation rate, the firn density profiles at Summit and WAIS Divide should be nearly identical. Additionally, the model should be able to replicate these profiles using these two meteorological conditions. The idealized density profile generated from the model appeared to capture the general densification trend with depth between 3.5–11 m at Summit, although it failed to capture the density variability in this portion of the profile. The model consistently underestimated firn density at WAIS Divide at depths between 3.5–11 m. Below, we investigate how other differences in meteorological conditions at Summit and WAIS Divide may result in the observed density discrepancies between profiles at Summit and WAIS Divide.

Air temperatures at Summit were characterized by more seasonal variability than those at WAIS Divide. Summer temperatures at Summit were higher than at WAIS Divide, though average winter temperatures were only slightly statistically different, with a small effect size (Table 1). Large seasonal temperature changes during summer can impose a large temperature gradient between the relatively cold snow and the warmer overlying air [29]. Temperature gradients imposed on surface snow from overlying air enhances vapor diffusion within the snowpack and drives snow metamorphism and densification [30]. The firn temperature at 10 m depth at Summit and WAIS Divide was −31.1 °C and −29.5 °C, respectively, suggesting that there was a greater heat flux in the WAIS Divide firn transferring these surface temperatures to greater depths [31,32]. Firn densification increases with increased temperature due to the Arrhenius-type dependence of normal grain growth and ice creep with temperature [12,33,34]. Vapor transport, following temperature and vapor pressure gradients into the colder snow and firn at depth during the summer, increases the density of snow and firn below the surface. This vapor transport can be reversed during winter [29]. Higher summer temperatures at Summit may increase firn compaction rates by driving vapor downwards into the firn and enhancing normal growth of large rounded grains.

In addition to greater seasonal variability in air temperatures, daily air temperatures fluctuated most at Summit, particularly during the spring (Figure 5). Rapid seasonal and diurnal variations in air temperatures can impose large temperature gradients conducive to forming coarse-grained, low-density hoar layers at or near the snow surface [34,35]. Seasonal fluctuations in temperature can

result in surface hoar forming in autumn when a rapid onset of cold temperatures cools the surface snow while the underlying layers are relatively warm [35]. Additionally, diurnal fluctuations in temperature can warm near-surface snow during the day, which can drive vapor upward out of the snowpack when the temperature gradient reverses at night, or vapor can move downward into colder layers [34]. These hoar layers, forming both at and near the surface, can be preserved within the firn column as low-density depth hoar layers and lower the bulk density of the firn column if these layers are buried by snow before being scoured by wind [35,36]. Further, the Summit site (3200 m a.s.l.) sits at a higher elevation than WAIS Divide (1776 m a.s.l.), causing higher sublimation rates due to lower air pressures. These higher sublimation rates promote hoar formation and less dense firn. Because of larger daily and seasonal contrasts in temperature, as well as the site elevation, we expect more surface and depth hoar to be present within the firn column at Summit, lowering the bulk density of this profile compared to WAIS Divide. We observed summit firn to be less dense than firn at WAIS Divide, confirming that hoar formation at this site likely reduces the overall density of the firn column.

Wind speeds at WAIS Divide consistently exceeded those at Summit, and WAIS Divide often experienced short-term periods of very high wind speeds (Figure 4). High wind speeds create rounder, finer surface snow crystals and increase the ability for these grains to be more closely packed [37]. Craven and Allison (1998) [38] found that, in Antarctica, higher wind speeds increase grain settling, the dominant driver of firn densification, up to a density of ~550 kg m$^{-3}$ [12,34]. The authors suggested that an increase in wind speed by 5 m s$^{-1}$ results in an equivalent increase in the firn densification rate as would be expected from a temperature increase of 10 °C [38]. Additionally, Fegyveresi and others (2018) [25] found that surface crusts often form during the austral summer months at WAIS Divide. Thus, we expect WAIS Divide to experience a higher degree of wind packing than Summit, resulting in denser firn.

Interestingly, although the bulk firn density differed between the two sites throughout the upper portion of the firn column and differences in firn density values persisted with depth, the firn permeability between 3.5–11 m depth was relatively similar (Figure 2) and the difference in the average bulk firn permeability was less statistically significant, with a relatively small effect size. This study provides more evidence that firn density is not necessarily a good indicator of firn permeability (as noted in [14,27,39]). Adolph and Albert noted that their best-fit parameterization based on open porosity and permeability is site-specific [14], and that open porosity is not a good indicator of permeability, especially in near-surface firn where permeability variability is high.

Herron and Langway developed their firn densification model to aid in identifying the pore close-off depth for interpreting gas measurements from ice cores [12]. Pore close-off depth at sites with higher accumulation rates typically occurs 50–80 m below the ice surface [34]. In many scenarios, the densification model has been shown to estimate pore close-off depth reasonably well [12,40]. However, firn densifies primarily through recrystallization and deformation below the 550 kg m$^{-3}$ density horizon. This process is sensitive to temperature, but below 10 m, diurnal and seasonal oscillations in air temperature are dampened and firn is approximately the same temperature as the mean annual air temperature. Thus, modeling densification processes at depths near the pore close-off using mean annual air temperature may be appropriate. However, the structure of firn in the upper portion of the firn column is affected by densification that varies based on changing meteorological conditions at the surface. As a result, using only two meteorological variables may not accurately determine the firn structure.

Modeling firn densification is essential for many applications, including the interpretation of ice core paleoclimate records. Knowing the depth at which firn reaches the critical close-off density, where gasses become trapped within bubbles in the ice, is critical to estimating the gas-age ice-age difference (Δage) in ice cores. Assuming that the pore close-off density is the same at both Summit and WAIS Divide, our study suggests that if the surface meteorology results in denser firn with depth at WAIS Divide, then the pore close-off depth will be shallower. This phenomenon has indeed been observed at WAIS Divide and Summit, where the pore close-off depth has been found to be 76.5 m and

80 m, respectively [23,41]. We would then expect the Δage to be different at these sites despite firn densification models predicting the same pore close-off depth. Therefore, sites like WAIS Divide, and any other site experiencing surface meteorological conditions outside of those observed at sites used to calibrate firn densification models, may have complicated structural differences in the firn column that impact gas movement through the open porosity and pore close-off processes.

Our study highlights the need to incorporate additional meteorological variables into models of firn structure. From our discussion above, we would expect firn at WAIS Divide to be denser given the site is less conducive to surface and depth hoar formation and that windy conditions create smaller, rounder grains that are more closely packed. Indeed, we found that firn was consistently denser at WAIS Divide than at Summit, but the Herron and Langway model failed to capture this difference. Furthermore, for depths between 3.5 m and 11 m, the firn density varied significantly between the sites, while the bulk firn permeability values were more similar. This result further stresses the need for a theoretical snow metamorphism model that generates accurate firn permeability profiles without relying on parameters derived from firn density.

## 5. Conclusions

We have shown that upper firn column densities at Summit and WAIS Divide are significantly different, despite the two sites having similar long-term meteorological characteristics. Firn between 3.5–11 m at Summit was consistently less dense than near-surface firn at WAIS Divide. We attribute density discrepancies to other average weather conditions at the two sites: Seasonal and daily variations in surface air temperature and annual differences in wind speed. At depths between 3.5 m and 11 m, while the difference in bulk firn permeability was statistically significant, a small effect size and similar median permeability values suggest that the bulk permeability is likely similar, and that firn density is not a good indication of permeability. Firn density and permeability are especially important for interpreting paleoclimate records, satellite altimetry, and predicting the available pore space for meltwater infiltration in increasingly warmer climates. The existing empirical firn densification models using mean annual air temperature and accumulation rate may fail to capture the densification of shallow firn. Furthermore, modeled permeability from observed or modeled density profiles may not generate an accurate permeability profile. This study suggests that physically based models incorporating additional meteorological variables should be used when modeling the structure of shallow firn.

**Author Contributions:** Conceptualization, M.R.A. and K.M.K.; Methodology, I.E.M., M.R.A.; S.A.L. and K.M.K.; Supervision, K.M.K.; Visualization, I.E.M.; Writing—original draft, I.E.M. and K.M.K.; Writing—review & editing, I.E.M., M.R.A., S.A.L. and K.M.K. All authors have read and agreed to the published version of the manuscript.

**Funding:** This research was funded by NSF Office of Polar Programs, grant numbers 0944078, 0520445, and 1443341.

**Acknowledgments:** The authors appreciate the support of the University of Wisconsin-Madison Automatic Weather Station Program for the dataset, NSF grant number 1924730. We thank the U.S. Ice Drilling Program for retrieving the Greenland and Antarctic firn cores. Dartmouth undergraduate students Emily Harwell, Julie Haldeman, Elinore Beitler, and Krystina Miles contributed to the cold room firn measurements. We thank the three anonymous reviewers for their helpful suggestions and comments that improved the manuscript.

**Conflicts of Interest:** The authors declare no conflict of interest.

## Appendix A

We examined the NCEP-NCAR reanalysis data [22] to place the WAIS Divide meteorological data in a larger temporal context and to ensure that our assumption that analyzing meteorological conditions in ten years after the firn core was collected would produce similar results to those we would have generated if we were able to analyze meteorological conditions over the period of time that influenced firn densification. We examine air temperature and relative humidity spanning the years 1995–2019 to determine whether meteorological conditions changed between the 10 years prior

to core collection in 2005, and the period of time from which we analyzed AWS meteorology data after core collection between 2009–2019.

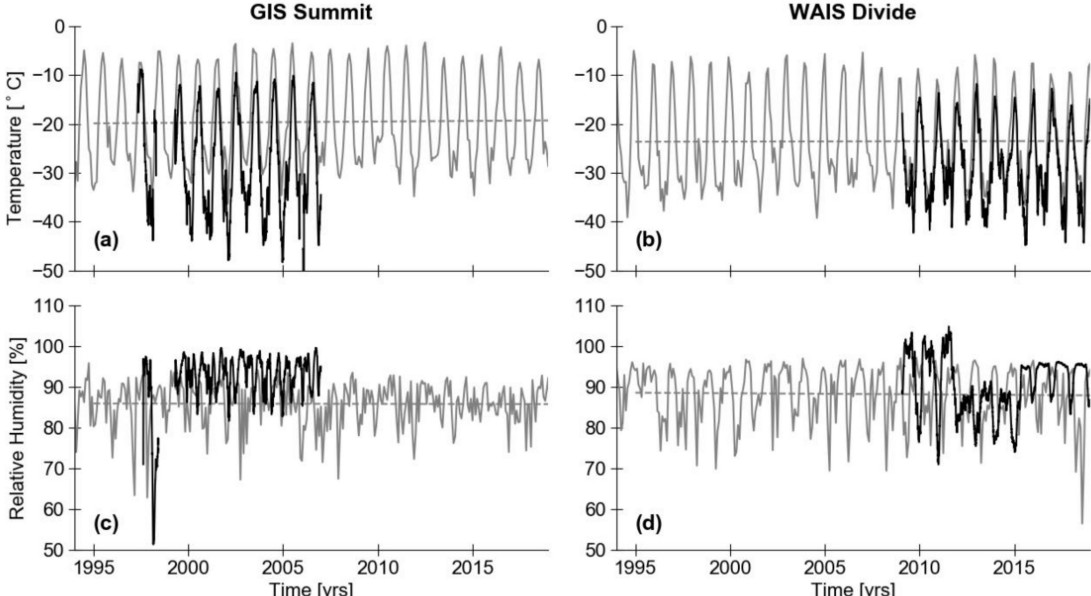

**Figure A1.** NCEP-NCAR reanalysis temperature and relative humidity (grey) plotted with 30-day running means of AWS observed variables (black) for Summit, Greenland (**a**,**c**) and WAIS Divide, Antarctica (**b**,**d**). Dashed grey lines show least-squares regression fits to the reanalysis data, showing that meteorological conditions remained essentially constant over the previous 25 years. Discrepancies between observed data and reanalysis data are attributed to local variability not captured in the regional nature of the 2.5° × 2.5° gridded reanalysis output.

We examined monthly mean values of the NCEP-NCAR reanalysis data from 2.5° × 2.5° gridded output covering both Summit and WAIS Divide. WAIS Divide data were downloaded from coordinates 79.5° S and 112.5° W from the 850 mb pressure level, corresponding closely with the air pressures recorded by the Kominko-Slade AWS. The AWS was a CR1000 model equipped with platinum resistance thermometers to measure temperature; Paroscientific, Druck, and Vaisala pressure transducers; and a Belfort wind sensor. Reanalysis data closely corresponding to Summit, were downloaded from the 700 mb pressure level centered at 72.5° N and 38.5° W, closely matching the AWS location and measured air pressures.

Reanalysis temperatures and relative humidity remain relatively constant at both Summit and WAIS Divide between 1995 and 2019 (Figure A1). At WAIS Divide, least-squares regressions fit to the reanalysis data have slopes of 0.006 °C yr$^{-1}$ and −0.027% yr$^{-1}$ for temperature and relative humidity respectively (Figure A1b,d). These trends give us confidence that the meteorological conditions have remained essentially constant between the period that would have influenced firn metamorphism of the WAIS Divide core and the temporal period over which we analyzed the AWS data.

At both Summit and WAIS Divide, visual discrepancies exist between the AWS data and reanalysis data. Observed temperatures at both sites are consistently lower than the reanalysis temperatures (Figure A1a,c). While the seasonal variations in temperature track closely, the reanalysis temperature sinusoid has been translated upwards by a few degrees. Additionally, relative humidity from the AWS at Summit is consistently higher than the NCEP-NCAR reanalysis relative humidity, while WAIS Divide has two distinct periods of higher relative humidity. Because we see discrepancies between reanalysis data and AWS data at both Summit and WAIS Divide, we attribute these differences to the inability for the reanalysis data to capture the local variability in meteorological conditions captured by the individual weather stations. We believe that our AWS data provides the best meteorological characterization of each site; however, the reanalysis data does give us confidence in the fact that

the AWS data we analyzed from a period after the WAIS Divide firn core collection would not be significantly different from the period that influenced snow deposition and firn metamorphism.

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
