# Peer review of "Local Weather Conditions Create Structural Differences between Shallow Firn Columns at Summit, Greenland and WAIS Divide, Antarctica"

_atmosphere, doi:10.3390/atmos11121370_

Round 1

Reviewer 1 Report

Review of "Local weather conditions create structural differences between shallow firn columns at Summit, Greenland and WAIS Divide, Antarctic" submitted to Remote Sensing journal by McDowell et al.

This is an interesting article about physical properties of firn cover at two important sites on West Antarctic and Greenland Ice Sheets. The article is well written and the presented results are compelling, nevertheless there are some issues that need to be resolved before it can be accepted for publication.

Unfortunately, the model of Adolph and Albert (2014) has not been properly introduced and the reader has to refer to the original paper, please provide a detailed description of it, including possible deficiencies and, most importantly, equations. I also suggest  comparing your measurements to some other firn model, I wonder how would that perform.

I think that you cannot directly compare hourly and 10-minute averages for GIS and WAIS sites,  the meteorological measurements have to be standardized and averaged over the same interval, otherwise you can misinterpret the short-term variability. Moreover, the simple assumption that the meteorological conditions at WAIS did not change for the period with no observation has to be more firmly supported, I strongly suggest using reanalysis data for this case. This could also help to place the meteorological record in a more long-term climatic context.

Finally, after reading the entire manuscript, I was left with a question - what impact your results might have for the interpretation of the climatic record in firn and ice cores, especially those deeper ones? Could you also provide a deeper discussion of your results with the outcome of studies from other sites?

Figure 1: I don;t like this figure, it could have been much better if you used sat imagery as a background; add grid and mention the coordinate system you used. Additionally, it seems to have very low resolution

Figure 2: add panel labels (a-f); I wonder, maybe it would have been much better for the reader if you changed your sites names? GIS Summit and WAIS Divide look way more clear to me, otherwise it is easy to mix them. Please add model uncertainty and some statistical metrics for a more descriptive figure.

L176-177: please introduce your error metrics once and then use it consistently, now the text is hard to follow if it is full of "(mean ± st. dev.)"

Figure 3: add panel labels; the distributions are far from gaussian; please explain why and support your choice of statistical measures (mean, std dev etc). Generally, this figure is very hard to follow and unclear, please add more labels and enhance its readability.

Figure 4: please add panel labels; following a major comment: it is hard to compare 10-minute and 1-hour averages, especially in case of wind speed, as it stands now the short-term variability is simply incomparable between WAIS and the Summit. How do you obtain RH>100%? Why is the lower bound of RH for WAIS divide filtered, approx at 45%?

Figure 5: please integrate all graphs into one panel

Table 1: why are there "-" signs before summer and winter air Temperature in the first column? The Table is badly formatted

Author Response

Thank you very much for your comments. Your suggestions have made this a better paper. Please see the attachment for a detailed response to your review.

Reviewer 2 Report

This is a wonderful and well written manuscript that reports on an inter comparison of firn profile properties between the two major ice sheets on earth: Greenland and Antarctica. The work is of high interest and is valuable to the community, because the results shed light on 3 major processes which mediate either understanding of, or impacts of, the firn layer on the larger world: (1) the ability of the firn to trap air, then used in ice-core interpretation, (2) the ability of firn to buffer meltwater runoff and thus impact sea level rise rates, (3) the problem that firn densification represents to the interpretation of satellite-altimetry data used widely to detect global ice-volume change.

I’ve noted only a few minor issues in the hyphenation and writing. Overall, the writing is outstanding.

The work is presented in a convincing manner and I had few if any questions that might be considered by the authors if they were to do minor revisions:

  • It might be worth adding to the description of the AWS stations the approximate heights of the instruments above the surface (of course these heights are changing, but worth mentioning).  Also, are there significant differences in solar SW radiation between the two sites, I assume not, but worth mentioning.
  • Is Relative Humidity calculated above water or above ice (the difference is probably irrelevant for the study, but worth noting, as if above water, the RH goes super 100% I think in really cold conditions).
  • In Fig 1. I suggest using low resolution of the Polar Geospatial Center DEM’s of Greenland and Antarctica instead of the gray outlines of the land masses… it would look classier…

Author Response

(The authors gave the same response as above.)

Reviewer 3 Report

McDowell et al. compare density and permeability of firn from 3.5-11 m depth from two sites with similar or even almost identical annual mean temperature and accumulation rate. Due to the much higher winds at WAIS Divide (WD), Antarctica, the density there is significantly higher than at Summit, Greenland. McDowell et al. point out that the permeability, generally parametrized by density, is similar at both sites but conclude as in a former study that density is not a good indicator of permeability in such shallow depths. Interesting is that both permeabilities are similar like annual mean temperature and accumulation rate despite the obvious differences in density (higher at WD), relative humidity (lower at WD), wind speed (higher at WD) and how temperature varies over the year.

The study presents convincing meteorological data to explain where both sites are similar and where they are different. I can accept the explanations given but I am not yet sure whether the explanations given for the similar permeability at the two sites do convince me. Firn is a complex medium/material and the meteorological/climatological boundary conditions can be quite diverse from year to year so the matter is also complex.This is probably not the last paper dealing with this topic. I like this little study but I also have some questions - see below.

Firn temperature is an important boundary condition for the metamorphism and the firnification processes in the firn column. You argue a lot with temperature related vapour fluxes and grain growth but present only air temperatures derived from the AWS-es. If firn temperatures are not available temperature in the firn could be modelled using the AWS data. Another hint for the firn temperatures could come from the 10 m firn temperatures although profiles would be preferred. Normally, both temperatures (AWS and 10m-firn) from the same sites show offsets. Do both sites have close/the same 10 m firn temperature as well? May be they are more different than the AWS temperatures? Are the temperature gradients similar/different? Heat conductivity is density related. The higher density at WD allows higher heat fluxes into/out of the firn. May be firn temperature helps explain why permeability at both sites is so close.

The paper is focussed on two sites only. So statistics is "poor". A question not discussed is whether there are other sites with similar meteorological conditions and similar density-depth relationship as Summit or WAIS Divide and what permeability such sites show. WAIS Divide has similar temperature and accumulation rate but the elevation is below 1800 m (WAIS 1776 m, Fudge GRL2016.pdf) and Summit above or about 3200 m above sea level. Is WAIS Divide an outlier and density can still be used to model permeability? Does WAIS Divide need special considerations? With the most experienced permeability expert amongst the co-authors ... you should have a few words for this aspect.

I am wondering that you do not or can not give a surface density for WD. Does it make sense to assume 350 kg/m3 for both sites? Known is that in the upper 2 m or so densification is not high if there is densification at all. Which processes in the uppermost 2-3 m lead to the much higher density at WD while at the same time permeability increases? How does this work?

Surprising is to see the large discrepancy in relative humidity at both sites. Knowing that the relative humidity at low temperatures can cause problems. Do you know whether both sites are equipped with the same sensors? Not only the differences appear surprising but also that WD has lower values during the austral winter. Often it is argued that rel. humidity increases when temperatures decrease. I do not question the low values at WD. Summit obviously shows values above 100%.

Statistics:
p-value and effective size. Please specify how you calculate the effective size.
I am not an expert on this topic. Most of your data are so clear so I want to ask what extra information p value and effective size contribute. If you want statistics bring into the game then data/profiles are needed. You could perhaps have a look into a recent paper by Weinhart about Surface density on the East Antarctic Plateau (https://doi.org/10.5194/tc-2020-14). You show just one model result. Have your varied boundary conditions for the models ? How sensitive is the model

Please add elevations of both site.

Figure 3: Please add number of samples

Figure 4: WAIS D has a changed y-axis scaling. Does the wind reach 60 m/s and more?

Grain growth: there is grain growth but it is normal (static) grain growth? Firn is a compacting material and you can not explain the increase in density without deformation I assume.

Author Response

(The authors gave the same response as above.)

Round 2

Reviewer 1 Report

I would like to thank authors for their effort to improve the manuscript, I believe now it is ready for publication, congratulations and with you good luck with future submissions!